# The Intention to Receive the COVID-19 Vaccine in China: Insights from Protection Motivation Theory

**DOI:** 10.3390/vaccines9050445

**Published:** 2021-05-02

**Authors:** Lu Li, Jian Wang, Stephen Nicholas, Elizabeth Maitland, Anli Leng, Rugang Liu

**Affiliations:** 1School of Business Administration, Jiangsu Vocational Institute of Commerce, Nanjing 211168, China; 200015@jvic.edu.cn; 2Dong Fureng Economic and Social Development School, Wuhan University, Beijing 100010, China; wangjian993@whu.edu.cn; 3Center for Health Economics and Management at School of Economics and Management, Wuhan University, Wuhan 430072, China; 4Australian National Institute of Management and Commerce, Eveleigh, Sydney, NSW 2015, Australia; stephen.nicholas@newcastle.edu.au; 5Research Institute for International Strategies, Guangdong University of Foreign Studies, Guangzhou 510420, China; 6School of Economics and School of Management, Tianjin Normal University, Tianjin 300074, China; 7Newcastle Business School, University of Newcastle, Newcastle, NSW 2308, Australia; 8School of Management, University of Liverpool, Chatham Building, Chatham Street, Liverpool L697ZH, UK; e.maitland@liverpool.ac.uk; 9School of Political Science and Public Administration, Institute of Governance, Shandong University, Qingdao 266237, China; lenganli@sdu.edu.cn; 10School of Health Policy & Management, Nanjing Medical University, Nanjing 211166, China; 11Center for Global Health, Nanjing Medical University, Nanjing 211166, China

**Keywords:** protection motivation theory, COVID-19, vaccination, intention, determinant

## Abstract

(1) Background: More coronavirus disease 2019 (COVID-19) vaccines are gradually being developed and marketed. Improving the vaccination intention will be the key to increasing the vaccination rate in the future; (2) Methods: A self-designed questionnaire was used to collect data on COVID-19 vaccination intentions, protection motivation and control variables. Pearson Chi-square test and multivariate ordered logistic regression models were specified to analyze the determinants of intention to receive COVID-19 vaccine; (3) Results: Although the vaccine was free, 17.75% of the 2377 respondents did not want, or were hesitant, to receive the COVID-19 vaccine. Respondents’ cognition of vaccine safety, external reward and response efficacy were positively related to COVID-19 vaccination intention, while age, income and response cost were negatively related to the intention to receive the COVID-19 vaccine. Professionals and people without medical insurance had the lowest intention to vaccinate; (4) Conclusions: The older aged, people without health insurance, those with higher incomes and professionals should be treated as the key intervention targets. Strengthening publicity and education about the safety and efficacy of COVID-19 vaccines, training vaccinated people and community leaders as propagandists for the vaccine, and improving the accessibility to the COVID-19 vaccine are recommended to improve COVID-19 vaccination intention.

## 1. Introduction

Corona Virus Disease 2019 (COVID-19) may be one of the most serious infectious diseases in human history, with over 119.2 million infected cases, and over 3 million deaths reported by World Health Organization as of 15 April 2021 [1]. COVID-19 vaccination programs covering the entire population will be the main way that countries control and prevent COVID-19. Vaccine hesitancy presents the major challenge for national vaccination campaigns, but the intention to COVID-19 vaccinate is poorly understood. High vaccine hesitancy rates, ranging from 14% to 30% have been variously reported in the United States of America [2], France [3], Japan [4], Canada [5] and the United Kingdom [6,7]. Even among healthcare professionals (HCP), the hesitancy rate for COVID-19 vaccinate has been reported at 24% in France, with only 64.4% HCP in Greece and 34.9% in the Republic of Cyprus intending to vaccinate [8,9]. Utilizing protection motivation theory (PMT), this paper analyzes the intention to COVID-19 vaccinate in China. PMT provides a widely used analytical framework to identify the determinants of vaccination behavior and vaccination willingness [10,11,12,13,14,15,16,17,18,19,20,21]. By providing a comprehensive theoretical framework on vaccination intentions, the paper not only reveals the factors determining the intention to vaccinate in China, but also informs policy makers on how to structure effective COVID-19 information campaigns to maximize the vaccine take-up.

Developed by Rogers in 1975 [22], protection motivation theory (PMT) argues that individuals are motivated to react in a self-protected way to perceived health threats. As shown in Figure 1, based on information, such as the environmental context and individual characteristics, the protection motivation decision involves two appraisal processes: threat appraisal and coping appraisal [22]. Threat appraisal depends on the individuals’ cognition of the threat, including the severity of the threat, one’s vulnerability, internal rewards and external rewards. Severity refers to people’s cognition of the magnitude of the harm caused by health hazards, such as COVID-19; vulnerability refers to individual’s perception of the possibility of suffering harm when they exposed to the health threat, such as the possibility of catching COVID-19; internal rewards refers to individual’s self-satisfaction after taking an action to protect from the health threat; external rewards refers to cost–benefit factors external to the individual, such as family, friends and the wider society of taking harm reduction through, for example, the COVID-19 vaccination [22]. The coping appraisal process evaluates an individual’s ability to cope with and avert the threatened health danger through self-efficacy, response efficacy and response cost [23]. Self-efficacy is the belief that one is, or is not, capable of harm reduction action, such as taking the COVID-19 vaccine [24]; response efficacy refers to one’s belief that taking a harm reduction action, such as COVID-19 vaccination, will be effective in health harm reduction; response cost are the barriers to taking protective behavior, measured by vaccine costs, vaccination knowledge and side-effect concerns.

PMT has been shown to be a powerful framework for behavior interpretation, intervention and prediction research on health harm reduction in China, the United States, Thailand, New Zealand, and Australia [10,11,12,13,14]. It has also been widely used to analyze the determinants of vaccination behavior and vaccination willingness [15,16,17]. For example, analyzing hepatitis B virus (HBV) vaccination behavior and willingness of Chinese migrant workers, Liu et al. found that vulnerability and response efficacy were significant determinants of HBV vaccination behavior, and vulnerability and self-efficacy were significant determinants of HBV vaccination willingness [18]. Among migrant workers in Tianjin, Liu et al. found that severity and self-efficacy were positively, and response cost negatively, related to HBV vaccination intention [19]. Ling et al. (2019) analyzed the intention to receive the seasonal influenza vaccine among 547 residents in the United States, and found that severity, susceptibility, the maladaptive response reward, self-efficacy, and response efficacy were unique determinants of vaccination intention, and the PMT factors explained 62% of the variance in intention to vaccinate [20]. A study in Switzerland showed that response efficacy was directly related to parents’ intention to adhere to measles, mumps and rubella (MMR) vaccination recommendations among parents of middle school students aged 13–15 [21]. Ling et al. used the PMT to predict the intention to receive the seasonal influenza vaccine, and found that severity of and susceptibility to influenza, the perceived benefits of not vaccinating, the self-efficacy to vaccinate, and the response efficacy were significant predictors of intention and the PMT variables accounted for 62% of the variance [20].

There has been a small number of COVID-19 PMT studies, including 734 healthcare workers in Iran [25], 649 Filipinos answering an online questionnaire [26], and 3145 students from 43 universities in China [27]. The online survey of Chinese students found that the perceived severity of COVID-19 was positively associated with motivation to have COVID-19 vaccination and receiving COVID-19 information from medical personnel was associated with greater self-efficacy, response efficacy, and knowledge, whereas receiving information concerning COVID-19 vaccination from coworkers/colleagues was associated with less response efficacy and knowledge. Although PMT has been proved to be a robust analytical framework to predict the vaccination intention and its determinants, there has been no nationwide study using PMT to analyze the determinant of residents’ intention to receive the COVID-19 vaccine in China. Covering all occupations, rural–urban and geographic regions, age, sex, income, education level and specifying a full PMT model, this paper analyzes factors determining the intention to vaccinate in China.

## 2. Materials and Methods

### 2.1. Sampling Method

Based on previous related research [18], a questionnaire was designed to collect information on COVID-19 vaccination intentions, PMT factors and control variables, such as age, sex, occupation and health. Nine provinces were selected randomly from China’s 27 provinces, with equal numbers from the eastern, central and western region and equal numbers of low, medium and high GDP based on based on each province’s 2019 GDP. Three cities were selected randomly from each province according to their 2019 GDP rank. All investigators were recruited from local colleges or universities and they received standardized training before commencing the investigation. Snowball sampling was applied with 100 participants interviewed face-to-face (or through online video for respondents required to home quarantine) in each city. The investigators were asked to choose respondents with equal numbers of males and females and urban and rural residents in a 3:2 ratio to reflect the national urban–rural distribution. The survey was conducted during the first two weeks of June 2020. All respondents were informed about the survey purpose and signed the informed consent form before the interview. A total of 2700 adults aged over 18 years old were interviewed. After deleting invalid cases with missing data, the final sample of 2377 respondents represented an 88.04% response rate.

### 2.2. Measurement of Intention to Receive COVID-19 Vaccine

The Chinese government has made the COVID-19 vaccine free, with high risk individuals receiving the vaccine first [28]. As of April 2021, over 190 million Chinese have received the same COVID-19 vaccine, with only 0.002% serious adverse reactions reported in Hong Kong and similar rates in mainland China [29]. The intention to receive the COVID-19 vaccine was measured by a one-item question: “Would you like to receive the COVID-19 vaccine if the vaccine is free?” with three answers, “No” (0), “it depends” (1) and “Yes” (2) representing the COVID-19 vaccination intention from low to high commitment.

### 2.3. Measurement of PMT Factors

Table 1 shows the one-item question to measure each PMT factor, with respondents replying on a three-point scale: “disagree” (0), “neutral” (1) and “agree” (2).

### 2.4. Measurement of Control Variables

The control variables included sex, age groups, average monthly income groups, education level, occupation, medical insurance, urban or rural residence, self-rated health level, east–west–central region, and respondents’ awareness of COVID-19 vaccine efficacy, safety and infection risk. Occupations were categorized into professionals (including physicians, teachers and civil servants), farmers, migrant workers, self-employed, unemployed, students, the retired and other. Self-rated health level was classified into, “bad”, “medium” and “good”, based on the question: “How is your health status compared to your peers?” A single-choice question was used to measure respondents’ awareness of vaccine safety, “Do you believe COVID-19 vaccine is safe?”, and coded “don’t agree—low safety”, “neutral attitude—medium safety” and “agree—high safety”.

### 2.5. Statistical Analyses

The database was built by using software EpiData 3.1 (The EpiData Association, Denmark), with all data double-entered and checked for consistency, and STATA 12.0 (StataCorp., College Station, Texas TX, USA) used for statistical analysis. Pearson chi-square test was used to compare the differences in COVID-19 vaccination intentions among subgroups and multivariate ordered logistic regression models, and the odds ratio (OR), were used to assess the associations between each independent variable and the COVID-19 vaccination intention. The underlying hypothesis of the study is that the control variables and PMT factors had significant influence (*p* < 0.05) on participants’ COVID-19 vaccination intention.

## 3. Results

### 3.1. Statistical Description of Respondents’ Characteristics and PMT Factors

Table 2 displays the characteristics of 2377 participants. The median age was 35 years old; broadly equal number of males (48.55%) and females (51.45%); the ratio of urban (61.51%) and rural (38.49%) respondents close to the Chinese national urban–rural 3:2 ratio. The median monthly income was RMB 5000, with the percentage in each income group broadly equal. Roughly two-thirds (61.93%) of respondents had a below high school education level; 29.9% were professionals, 26.88% students, 12.16% migrant workers; 96.93% of participants reported their self-rated health as “medium or good”; only 3.7% had no medical insurance; the distribution across eastern (31.47%) central (28.86%) and western (39.67%) regions were roughly equal. Most participants believed the COVID-19 vaccine was safe at a medium or high level (82.41%) and overwhelmingly agreed that COVID-19 was a serious disease (91.67%). A quarter of respondents believed that they, their relatives and friends could be infected by COVID-19; 46.87% believed that they would not be restricted in their travel by COVID-19 after receiving the vaccine; 63.4% believed their relatives and friends would receive COVID-19 vaccine. Most respondents believed that they had the ability to receive the COVID-19 vaccine (77.96%) and that the COVID-19 vaccine would be effective against COVID-19 (86.62%). Only 14.51% believed going to receive the COVID-19 vaccine would waste time and delay their work. As shown in Figure 2, 82.25% intended to receive the COVID-19 vaccine, but 14.05% were hesitant to vaccinate and 3.7% did not want to get vaccinated.

### 3.2. Results of Pearson Chi-Square Test

Table 2 illustrates the intention to receive COVID-19 vaccine by subgroups. The percentage of people who did not want the COVID-19 vaccine was highest in the over 58 years old group (7.2%) and lowest in the 18–27 years old group (2.3%) (*p* = 0.001). The COVID-19 vaccination intention rate decreased with rising income, with the lowest vaccination intention in the highest income group (78.97%) (*p* = 0.004). Respondents with the lowest education level (lower than high school education) had a significantly higher intention (83.42%) to receive the COVID-19 vaccine than respondents who had a high school and above education (80.33%, *p* = 0.021). Among all occupations, professionals had the lowest intention (78.95%) and farmers had the highest intention (88.85%) to COVID-19 vaccinate (*p* = 0.001). The respondents who had medical insurance (82.66%) had a higher intention of vaccinating than the noninsured respondents (71.59%, *p* = 0.015). Respondents from the central region (83.53%) had the highest intention to revive the COVID-19 vaccine, followed by the western area (82.61%) and then the more affluent eastern area (80.61%, *p* = 0.034). The vaccination intention increased significantly with participants’ increased awareness of vaccine safety (*p* < 0.001).

There were significant differences across all PMT subgroups (*p* < 0.05). The percentage of respondents who did not want to receive COVID-19 vaccine decreased with an increased level of severity, internal reward, external reward, self-efficacy and response efficacy, but increased with increased response cost level (*p* < 0.001). Those participants who agreed that they and people around them might get COVID-19 in the future had the highest intention to receive the COVID-19 vaccine (84.33%), followed by the disagree group (82.57%) and neutral group (79.85%) (*p* = 0.013).

### 3.3. Results of Multivariate Ordered Logistic Regressions

Table 3 shows the results of multivariate ordered logistic regression models, specified to analyze the relationship between intention to receive the COVID-19 vaccine and the independent variables. The independent variables in Model 1 comprised only the control variables, while Model 2 specified both the PMT factors and the control variables. The value of log likelihood and pseudo R^2^ in Model 2 (Log likelihood = −1116.2274, Pseudo R^2^ = 0.1592) were lager than that in Model 1 (Log likelihood = −1203.4388, Pseudo R^2^ = 0.0935), so Model 2, with the PMT factors, performed better at explaining the intention to receive the COVID-19 vaccine.

Among the control variables, sex, education level, urban–rural residence, self-rated health and region had no influence on respondents’ intention to receive the COVID-19 vaccine (*p* > 0.05). Respondents in all age groups, except those 28–37 age years old, had a higher intention to receive COVID-19 vaccine than the 18–27 age group, and the intention to vaccinate decreased as age increased (OR = 0.637 (age 38–47), OR = 0.594 (age 48–57), OR = 0.471 (age 58+), *p* < 0.05). Respondents in the low income group had a higher intention to vaccinate than the high income group (OR = 1.705, *p* = 0.002); farmers had a higher intention to receive the COVID-19 vaccine than professionals (OR = 2.134, *p* = 0.005); those without medical insurance had a lower COVID-19 vaccination intention than the insured (OR = 0.584, *p* = 0.049). The intention to vaccinate was higher by respondents who perceived the COVID-19 vaccine had high safety than respondents who perceived the vaccine had a low safety (OR = 0.546, *p* < 0.001).

Among the PMT variables in Model 2, severity, vulnerability, internal reward and self-efficacy had no influence on respondents’ intention to receive the COVID-19 vaccine (*p* > 0.05). There was a significant association between intention to receive the COVID-19 vaccine and external reward, response efficacy and response cost (*p* < 0.05). The respondents who agree with the external reward description were more likely to receive the COVID-19 vaccine than respondents who disagree (OR = 4.519, *p* < 0.001). The vaccination intention increased as response efficacy increased (neutral: OR = 3.105, *p* = 0.041; agree: OR = 5.768, *p* = 0.001), but deceased as response cost increased (neutral: OR = 0.749, *p* = 0.047; agree: OR = 0.5, *p* < 0.001).

## 4. Discussion

Based on protection motivation theory, the intention to receive the COVID-19 vaccine and its influencing factors were analyzed. The outcomes can guide COVID-19 vaccine uptake and shape vaccination policy. The COVID-19 vaccination intention rate was 82.25%, but 17.75% of respondents did not want, or hesitated, to receive the free COVID-19 vaccine. The vaccination intention rate was higher than that reported in Hong Kong (from 34.8% to 44.2%) and in an online survey in China (54.6%) [30,31]. Lower COVID-19 vaccination rates also were reported in other Asian countries with 65.7% of participants indicating a willingness to be vaccinated against COVID-19 in Japan [4], 53.1% in Kuwait [32], 78.3% in Indonesia [33] and 64.7% in Saudi Arabia [34]. A multi-country survey in Europe showed on average 73.9% respondents from Germany, the United Kingdom, Denmark, the Netherlands, France, Portugal and Italy were willing to receive the COVID-19 vaccine. In North America, over half of Canadians were very likely (57.5%) to get a COVID-19 vaccine when it becomes available and 19.0% reported that they were somewhat likely to get vaccinated [5]. A online survey in the United States showed that 69% of participants intent to get vaccinated against COVID-19 [35]. In South America, 90.6% of participants indicated they were willing to pay for a COVID-19 vaccine in Chile [36], and a survey in Ecuador showed that at least 97% of individuals were willing to accept a COVID-19 vaccine, and at least 85% were willing to pay for that vaccine [37]. In Australia, 85.8% of respondents would get the COVID-19 vaccine [38], but in the Congo only 27.7% of health care workers said that they would accept a COVID-19 vaccine [39].

Age, income, occupation, medical insurance, vaccine safety, external reward, response efficacy and response cost were significant influencing factors of the intention to receive the COVID-19 vaccine. Sex, education level, urban–rural residence, self-rated health, region, severity, vulnerability, internal reward and self-efficacy were not significant factors in the intention to vaccinate.

For the agree to vaccinate group, the most important determinants to vaccinate was PMT response efficacy (OR = 5.768), followed by vaccine safety (OR = 5.546 for the high safety level group) and external reward (OR = 4.519 for the agree group). These results were consistent with previous studies which found respondents’ perception of the potential risk and harm of the COVID-19 vaccine decreased the intention to get vaccinated [32,35], while the perceived effectiveness of the vaccine increased vaccination intention [30,40,41,42]. Previous research on HBV [43], MMR [21] and seasonal influenza [20] vaccination intentions also found that response efficacy was a unique determinant of vaccination intention. To increase the vaccination rate, a publicity and education campaign about the safety and efficacy of COVID-19 vaccines would strength public trust in COVID-19 vaccines, increasing vaccination intentions [44,45,46]. Since urban–rural residence and region had no influence on respondents’ vaccination intention, education campaigns should be national. News, even negative news, about COVID-19 vaccines should be reported accurately, not exaggerated or sensationalized. External reward indicated that respondents’ intention to receive the COVID-19 vaccine was influenced by the viewpoint of family members, relatives and friends. We recommend using respected members of the community, such as community leaders or professionals, as well as vaccinated people, as propagandists for vaccinations to influence the population to vaccinate. Response cost indicated that the increased cost of vaccination, such as lost work time and travel time, decreased the intention to get COVID-19 vaccinated. This finding was similar to previous studies of HBV vaccination intention, which found that response cost was negatively related to intention to receive the HBV vaccine [19]. We recommended that improving the accessibility to the COVID-19 vaccinate to increase the population’s vaccination intention, such as increasing the number of vaccination sites to reduce commuting time.

Even though the COVID-19 vaccine was free, the intention to COVID-19 vaccinate for respondents without medical insurance was only half those with medical insurance. Other COVID-19 research also found that vaccine acceptance was higher among the U.S. private or public health insured than those without health insurance [35]. The effect of age, income and occupation on intention to COVID-19 vaccinate was different from other studies. We found that the intention to COVID-19 vaccinate decreased with age, except for the 28–37 years old age group. In contrast, Japanese and U.S. studies found that the older age respondents had a higher intention to COVID-19 vaccinate than younger people [4,47]. In part, this can be explained by the evidence that the likelihood of COVID-19 infection and mortality from COVID-19 increased with age [48], so measures need to be implemented to increase the COVID-19 vaccination intention of the older aged. Our model showed that professionals had the lowest intention to COVID-19 vaccinate, while farmers had the highest vaccination intention. Additionally, different from other studies [33,37,49] was that respondents with lower income had a higher intention to receive COVID-19 vaccine than respondents in the high income group. The different effect of income and occupation on COVID-19 vaccination intention might reflect that the COVID-19 vaccination was free. Respondents with lower socioeconomic status are more sensitive to vaccination cost, so are more willing to vaccinate. We recommend that the older aged, people without health insurance, those with higher incomes and professionals should be the key intervention targets to improve vaccination intentions.

### Strengths and Limitations

The strengths of this study are as follows. First, we used PMT, which provides a robust, well-tested and full model to explain the cognitive mediation process of behavioral change in terms of threat and coping appraisal. In a nationwide study, our PMT model analyzed the determinants of intention to receive the COVID-19 vaccine with all seven PMT factors included. Second, COVID-19 vaccination intentions were analyzed under China’s free vaccine context. Third, the data were collected from the eastern, central and western regions as well as rural and urban areas. Lastly, respondents were interviewed face-to-face or through online video, which enhanced data accuracy compared to online questionnaire surveys.

There are two major limitations. First, only one question was set to evaluate respondents’ intention to receive the COVID-19 vaccine and each PMT factor. A more complex measurement method for PMT factors should be developed in further studies. Second, future studies should collect larger datasets.

## 5. Conclusions

Our PMT model found that 18% of respondents did not want, or were hesitated to get, the free COVID-19 vaccination. There were significant correlations between key control variables (age, income, occupation, medical insurance, and vaccine safety), PMT factors (external reward, response efficacy and response cost) and respondents’ intention to COVID-19 vaccinate. Perception of the vaccine’s safety, external reward and response efficacy were positively related to COVID-19 vaccination intention, while age, income and response cost were negatively related to the intention to receive the COVID-19 vaccine. Professionals had the lowest COVID-19 vaccination intention among all occupations, and people without medical insurance had a lower intention to vaccinate than those with medical insurance. When the COVID-19 vaccine is free, strengthening publicity and education about the safety and efficacy of COVID-19 vaccines, for example using vaccinated respected leaders as propagandists, and improving the accessibility to COVID-19 vaccination centers are recommended actions to improve COVID-19 vaccination intentions. The older aged, people without health insurance, those with higher incomes and professionals should be the key intervention targets.

## Figures and Tables

**Figure 1 vaccines-09-00445-f001:**
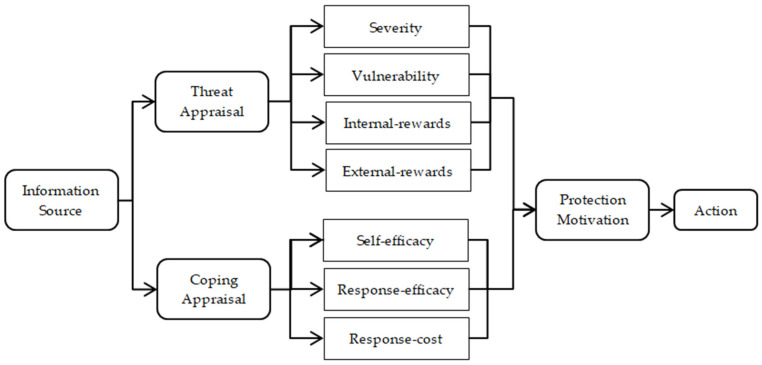
Protection motivation theory.

**Figure 2 vaccines-09-00445-f002:**
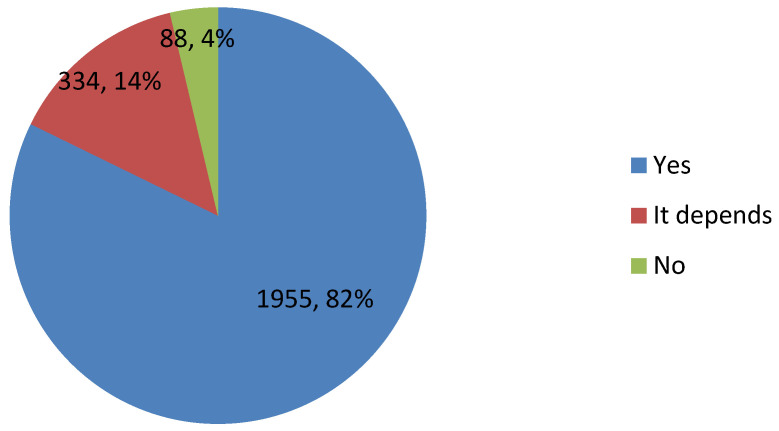
Overall percentage of intention to receive COVID-19 vaccine (number, percentage).

**Table 1 vaccines-09-00445-t001:** Measurement of PMT factors.

Factors	Description	Assignment
Severity	COVID-19 is a serious disease.	Disagree = 0; Neutral = 1; Agree = 2
Vulnerability	My relatives, friends and I face the risk of COVID-19 infection.	Disagree = 0; Neutral = 1; Agree = 2
Internal reward	After I received the COVID-19 vaccine, I will no longer be restricted in my travel.	Disagree = 0; Neutral = 1; Agree = 2
External reward	My relatives, friends and people around me all want to get vaccinated against COVID-19.	Disagree = 0; Neutral = 1; Agree = 2
Self-efficacy	I believe I will have the ability to get the COVID-19 vaccine in the future.	Disagree = 0; Neutral = 1; Agree = 2
Response efficacy	The COVID-19 vaccine is effective against COVID-19.	Disagree = 0; Neutral = 1; Agree = 2
Response cost	Going to get the COVID-19 vaccine would waste my time or delay my work.	Disagree = 0; Neutral = 1; Agree = 2

**Table 2 vaccines-09-00445-t002:** PMT factors and control variables.

Variables			Total		No		It Depends		Yes		χ^2^	*p*
			N	%	N	%	N	%	N	%		
PMT factors	Severity	Disagree	46	1.94	6	13.04	4	8.70	36	78.26	28.303	<0.001
		Neutral	152	6.39	6	3.95	38	25.00	108	71.05		
		Agree	2179	91.67	76	3.49	292	13.40	1811	83.11		
	Vulnerability	Disagree	1107	46.57	50	4.52	143	12.92	914	82.57	12.692	0.013
		Neutral	670	28.19	18	2.69	117	17.46	535	79.85		
		Agree	600	25.24	20	3.33	74	12.33	506	84.33		
	Internal rewards	Disagree	489	20.57	22	4.50	74	15.13	393	80.37	39.773	<0.001
		Neutral	774	32.56	33	4.26	150	19.38	591	76.36		
		Agree	1114	46.87	33	2.96	110	9.87	971	87.16		
	External rewards	Disagree	141	5.93	14	9.93	36	25.53	91	64.54	209.583	<0.001
		Neutral	729	30.67	46	6.31	187	25.65	496	68.04		
		Agree	1507	63.40	28	1.86	111	7.37	1368	90.78		
	Self-efficacy	Disagree	87	3.66	7	8.05	13	14.94	67	77.01	66.452	<0.001
		Neutral	437	18.38	26	5.95	107	24.49	304	69.57		
		Agree	1853	77.96	55	2.97	214	11.55	1584	85.48		
	Response efficacy	Disagree	17	0.72	6	35.29	4	23.53	7	41.18	245.711	<0.001
		Neutral	301	12.66	28	9.30	111	36.88	162	53.82		
		Agree	2059	86.62	54	2.62	219	10.64	1786	86.74		
	Response cost	Disagree	1500	63.10	52	3.47	166	11.07	1282	85.47	40.708	<0.001
		Neutral	532	22.38	20	3.76	117	21.99	395	74.25		
		Agree	345	14.51	16	4.64	51	14.78	278	80.58		
Control variables	Sex	Male	1154	48.55	45	3.90	156	13.52	953	82.58	0.720	0.698
		Female	1223	51.45	43	3.52	178	14.55	1002	81.93		
	Age	18–27	915	38.49	21	2.30	139	15.19	755	82.51	25.116	0.001
		28–37	340	14.30	12	3.53	37	10.88	291	85.59		
		38–47	460	19.35	21	4.57	77	16.74	362	78.70		
		48–57	426	17.92	17	3.99	60	14.08	349	81.92		
		>58	236	9.93	17	7.20	21	8.90	198	83.90		
	Income	High	813	34.20	33	4.06	138	16.97	642	78.97	15.355	0.004
		Medium	796	33.49	29	3.64	116	14.57	651	81.78		
		Low	768	32.31	26	3.39	80	10.42	662	86.20		
	Education level	Below high school	1472	61.93	59	4.01	185	12.57	1228	83.42	7.684	0.021
		High school and above	905	38.07	29	3.20	149	16.46	727	80.33		
	Occupation	Professional	708	29.79	29	4.10	120	16.95	559	78.95	35.573	0.001
		Farmer	278	11.70	9	3.24	22	7.91	247	88.85		
		Migrant worker	289	12.16	8	2.77	42	14.53	239	82.70		
		Self-employed	221	9.30	12	5.43	29	13.12	180	81.45		
		Unemployed	103	4.33	8	7.77	6	5.83	89	86.41		
		Student	639	26.88	14	2.19	100	15.65	525	82.16		
		Retired	86	3.62	4	4.65	9	10.47	73	84.88		
		Other	53	2.23	4	7.55	6	11.32	43	81.13		
	Medical insurance	Yes	2289	96.30	81	3.54	316	13.81	1892	82.66	8.392	0.015
		No	88	3.70	7	7.95	18	20.45	63	71.59		
	Residence	Urban	1462	61.51	56	3.83	209	14.30	1197	81.87	0.394	0.821
		Rural	915	38.49	32	3.50	125	13.66	758	82.84		
	Self-rated health	Bad	73	3.07	3	4.11	10	13.70	60	82.19	1.171	0.883
		Medium	564	23.73	24	4.26	74	13.12	466	82.62		
		Good	1740	73.20	61	3.51	250	14.37	1429	82.13		
	Region	Eastern	748	31.47	23	3.07	122	16.31	603	80.61	10.398	0.034
		Central	686	28.86	34	4.96	79	11.52	573	83.53		
		Western	943	39.67	31	3.29	133	14.10	779	82.61		
	Vaccine safety	Low	40	1.68	11	27.50	8	20.00	21	52.50	279.622	<0.001
		Medium	378	15.90	36	9.52	128	33.86	214	56.61		
		High	1959	82.41	41	2.09	198	10.11	1720	87.80		

**Table 3 vaccines-09-00445-t003:** Multiple ordered logistic regression models.

Variables			Model 1				Model 2			
			β	S.E.	*p*	OR (95%CI)	β	S.E.	*p*	OR (95%CI)
PMT factors	Severity	Disagree					(Reference group)			
		Neutral					0.681	0.45	0.130	1.976 (0.819, 4.767)
		Agree					0.674	0.40	0.090	1.962 (0.900, 4.276)
	Vulnerability	Disagree					(Reference group)			
		Neutral					0.161	0.15	0.269	1.175 (0.883, 1.565)
		Agree					−0.073	0.15	0.637	0.930 (0.686, 1.259)
	Internal rewards	Disagree					(Reference group)			
		Neutral					0.171	0.17	0.300	1.186 (0.859, 1.640)
		Agree					0.006	0.17	0.973	1.006 (0.725, 1.394)
	External rewards	Disagree					(Reference group)			
		Neutral					0.192	0.21	0.364	1.211 (0.801, 1.831)
		Agree					1.508	0.22	<0.001	4.519 (2.914, 7.009)
	Self-efficacy	Disagree					(Reference group)			
		Neutral					−0.309	0.32	0.330	0.734 (0.394, 1.367)
		Agree					−0.008	0.30	0.979	0.992 (0.547, 1.798)
	Response efficacy	Disagree					(Reference group)			
		Neutral					1.133	0.55	0.041	3.105 (1.048, 9.197)
		Agree					1.752	0.55	0.001	5.768 (1.956, 17.010)
	Reaction cost	Disagree					(Reference group)			
		Neutral					−0.288	0.15	0.047	0.749 (0.564, 0.996)
		Agree					−0.694	0.18	<0.001	0.500 (0.354, 0.705)
Control variables	Sex	Male	(Reference group)				(Reference group)			
		Female	−0.102	0.11	0.375	0.903 (0.721, 1.131)	−0.135	0.12	0.263	0.874 (0.690, 1.106)
	Age	18-27	(Reference group)				(Reference group)			
		28-37	0.116	0.23	0.621	1.123 (0.709, 1.779)	0.133	0.24	0.585	1.142 (0.709, 1.840)
		38-47	−0.367	0.21	0.083	0.692 (0.457, 1.049)	−0.451	0.22	0.040	0.637 (0.414, 0.979)
		48-57	−0.397	0.22	0.076	0.672 (0.434, 1.042)	−0.521	0.23	0.024	0.594 (0.377, 0.934)
		58-	−0.786	0.29	0.006	0.456 (0.260, 0.797)	−0.753	0.30	0.012	0.471 (0.263, 0.845)
	Income	High	(Reference group)				(Reference group)			
		Medium	0.192	0.14	0.169	1.212 (0.921, 1.594)	0.252	0.15	0.085	1.286 (0.966, 1.712)
		Low	0.434	0.16	0.008	1.544 (1.121, 2.126)	0.533	0.17	0.002	1.705 (1.218, 2.386)
	Education level	Below high school	(Reference group)				(Reference group)			
		High school and above	−0.202	0.14	0.151	0.817 (0.620, 1.076)	−0.203	0.15	0.165	0.816 (0.612, 1.088)
	Occupation	Professional	(Reference group)				(Reference group)			
		Farmer	0.706	0.26	0.007	2.026 (1.212, 3.386)	0.758	0.27	0.005	2.134 (1.258, 3.622)
		Migrant worker	0.099	0.21	0.632	1.104 (0.736, 1.655)	0.194	0.22	0.376	1.214 (0.791, 1.863)
		Self-employed	0.025	0.22	0.907	1.026 (0.671, 1.569)	0.121	0.23	0.594	1.129 (0.723, 1.760)
		Unemployed	0.621	0.35	0.076	1.860 (0.938, 3.691)	0.652	0.36	0.072	1.919 (0.944, 3.899)
		Student	−0.091	0.21	0.667	0.913 (0.603, 1.382)	−0.041	0.22	0.850	0.960 (0.625, 1.473)
		Retired	0.554	0.36	0.126	1.740 (0.856, 3.538)	0.651	0.38	0.084	1.917 (0.917, 4.008)
		Other	−0.041	0.39	0.916	0.960 (0.448, 2.055)	0.132	0.41	0.746	1.141 (0.514, 2.534)
	Medical insurance	Yes	(Reference group)				(Reference group)			
		No	−0.662	0.26	0.011	0.516 (0.311, 0.857)	−0.538	0.27	0.049	0.584 (0.341, 0.998)
	Residence	Urban	(Reference group)				(Reference group)			
		Rural	−0.096	0.13	0.451	0.909 (0.709, 1.165)	−0.151	0.13	0.253	0.860 (0.664, 1.114)
	Self-rated health	Bad	(Reference group)				(Reference group)			
		Medium	0.327	0.35	0.356	1.387 (0.692, 2.781)	0.666	0.36	0.066	1.947 (0.958, 3.957)
		Good	0.281	0.35	0.418	1.325 (0.671, 2.617)	0.507	0.35	0.152	1.66 (0.830, 3.322)
	Region	Eastern	(Reference group)				(Reference group)			
		Central	0.083	0.15	0.587	1.086 (0.806, 1.463)	0.092	0.16	0.563	1.096 (0.804, 1.494)
		Western	−0.039	0.14	0.779	0.962 (0.731, 1.265)	−0.025	0.15	0.867	0.976 (0.732, 1.300)
	Vaccine safety	Low	(Reference group)				(Reference group)			
		Medium	0.689	0.34	0.043	1.992 (1.023, 3.878)	0.695	0.37	0.063	2.004 (0.962, 4.173)
		High	2.400	0.33	<0.001	11.023 (5.731, 21.201)	1.713	0.37	<0.001	5.546 (2.688, 11.442)

## Data Availability

The data presented in this study are available on request from the corresponding author. The data are not publicly available due to multi-cooperation with Wuhan University, Shandong University and Nanjing Medical University. The corresponding author will facilitate a discussion with these three universities for data access on a reasonable request.

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
