# Peer review of "The Intention to Receive the COVID-19 Vaccine in China: Insights from Protection Motivation Theory"

_vaccines, 2021, doi:10.3390/vaccines9050445_

Round 1
Reviewer 1 Report
In this study authors present some interesting results based on a protection motivation theory model on the intention to vaccinate in a population in 27 Chinese provinces.
The results show that a total of 17.75% of those surveyed would not want to be vaccinated. The statistical study shows that age and income are negatively related. The older aged, people without health insurance and those with higher incomes and professionals should be treated as the key intervention targets.
This topic is interesting based in the growing need to improve the vaccination intention that will be the key to increase the vacunation rate probably in the most serious infectious disease in human history as authors refer
Abstract
Well written and well structured in Background, Methods, Results and Conclusion.
Introduction
The introduction explain clearly the concept of the Protection Motivation Theory (PMT).
Material and Method
Data analysis is correct using Person chi-square test as well that a multivariate logistic regressionmodel and oddds ratio.
However a study of the sample calculation would be necessary to show that a sample of 2377 respondents is significant
Results
The results are supported by two tables which clarifies their compression and despite their complexity they are summarized with good understandingDiscussion.
Author compared the vaccination intention rates in some countries with variable results.One of the biggest problems to reject the vaccine in Europe is in the type of vaccine and its relationship with thrombosis.
The same type of vaccine has been used in China for all cases ? It should be discussed what influence the publication of adverse effects may have on the acceptance of vaccination References The bibliography is complete and up-to-dateAuthor Response
In this study authors present some interesting results based on a protection motivation theory model on the intention to vaccinate in a population in 27 Chinese provinces.
The results show that a total of 17.75% of those surveyed would not want to be vaccinated. The statistical study shows that age and income are negatively related. The older aged, people without health insurance and those with higher incomes and professionals should be treated as the key intervention targets.
This topic is interesting based in the growing need to improve the vaccination intention that will be the key to increase the vacunation rate probably in the most serious infectious disease in human history as authors refer
Abstract
Well written and well structured in Background, Methods, Results and Conclusion.
Introduction
The introduction explain clearly the concept of the Protection Motivation Theory (PMT).
Material and Method
Data analysis is correct using Person chi-square test as well that a multivariate logistic regressionmodel and oddds ratio.
However a study of the sample calculation would be necessary to show that a sample of 2377 respondents is significant
Authors’ response:
Following previous research [Peduzzi P, Concato J, Kemper E, Holford TR, Feinstein AR. A simulation study of the number of events per variable in logistic regression analysis. J Clin Epidemiol. 1996 Dec;49(12):1373-9], we calculated the minimum sample size. The events per variable value usually is set to 10, so with 54 dummy variables in our regression model, minimum sample size should be 540÷82.25% (our response rate)≈657. Our sample was 2377.
Results
The results are supported by two tables which clarifies their compression and despite their complexity they are summarized with good understanding
Discussion.
Author compared the vaccination intention rates in some countries with variable results.One of the biggest problems to reject the vaccine in Europe is in the type of vaccine and its relationship with thrombosis.
The same type of vaccine has been used in China for all cases ? It should be discussed what influence the publication of adverse effects may have on the acceptance of vaccination
Authors’ response:
Thanks for your comments. Currently, 192 million Chinese have received the same free COVID-19 vaccine as of April 2021, with only 0.002% serious adverse reactions reported in Hong Kong and no consistent serious side effects reported in mainland China. Please see ‘Discussion’, Lines 142-144.
References The bibliography is complete and up-to-date
Reviewer 2 Report
This paper presents the results of a cross-sectional study of general public in Chan, related to Covid- 19 vaccination uptake and related factors, using protection motivation theory. Even new information is identified, however, the findings is weakened by several issues including incomplete description of background, method and results, inconsistency of gap identified, results and discussion, and issues in the logical presentation of the study aims. Details relating to these and other issues are presented below.
Overall:
- the authors conducted a comprehensive literature search to examine protection motivation theory, yet I am uncertain whether the study results are presented in a clear manner.
- Because of the organization of the manuscript, it is difficult to follow the logical flow; therefore, following items should be revised:
- paragraphs needs to be broken down by sub-topics/arguments.
- long sentences should also be shortened.
- the current numbering system for the result section needs to be revised.
- the discussion section should contain clear interpretation and implications of the study results.
Title
- Suggest: revise the title: Title does not match research purpose and content.
Introduction
Introduction:
- paragraphs should be broken down according to the sub-arguments
- the paragraphs should be re-organized to display following topics more logically:
- the importance/significance of this particular topic
- what has been studied about the topic
- research gap
- therefore what this study aims to achieve
- line 50-64: provide prefer references
Materials and Methods:
- Authors should revise methods:
- Regarding application of protection motivation model for intention to take Covid 19 vaccine of general public, more relevant articles should be included: especially studies on “uptake of influenza vaccine” and related variables, and Asian countries and religious and cultural issues.
- Authors should state the reasons for applying protection motivation theory to uptake of Covid -19 vaccination and related factors (why excluding safety? Only financing BARRIER?) with reviewing previous literature or observation. Additionally, authors should revise the last sentence…aims of this study...
- Authors should revise the method section carefully and rewrite following section more logically to better understand the text: study design, setting, Sample size and sampling method, Inclusion and exclusion criteria.
- provide “how to determine calculate the sample size for this study.
- provide the details of survey instrument: how to develop the survey questionnaire; to examine content and face validation of the survey instrument, variables of this study, and etc.
- Data collection
- provide details of: who are surveyors; informed consent; response rate; study periods, etc.
- Data analysis
- provide details of: statistical analyses for this study. (rely on hypothesis?)
- Result and Discussion
- Authors should revise the result and discussion section: considering appropriate existing and updated articles related to “protection motivation theory and vaccination”.
- References
- Authors should include more articles related to applying protection motivation theory and related factors.
Author Response
Reviwer#2
This paper presents the results of a cross-sectional study of general public in Chan, related to Covid- 19 vaccination uptake and related factors, using protection motivation theory. Even new information is identified, however, the findings is weakened by several issues including incomplete description of background, method and results, inconsistency of gap identified, results and discussion, and issues in the logical presentation of the study aims. Details relating to these and other issues are presented below.
Overall:
- the authors conducted a comprehensive literature search to examine protection motivation theory, yet I am uncertain whether the study results are presented in a clear manner.
- Because of the organization of the manuscript, it is difficult to follow the logical flow; therefore, following items should be revised:
- paragraphs needs to be broken down by sub-topics/arguments.
- long sentences should also be shortened.
- the current numbering system for the result section needs to be revised.
- the discussion section should contain clear interpretation and implications of the study results.
Title
- Suggest: revise the title: Title does not match research purpose and content.
Introduction
Authors’ response:
Following the reviewer’s suggestions, we have revised the title (Line 2-3).
Introduction:
2. paragraphs should be broken down according to the sub-arguments
3. the paragraphs should be re-organized to display following topics more logically:
- the importance/significance of this particular topic
- what has been studied about the topic
- research gap
- therefore what this study aims to achieve
Authors’ response:
We have substantially revised and re-organized the Introduction section (Lines 44-120), clearly identifying the significance of the study, the research gap and research aims.
4. line 50-64: provide prefer references
Authors’ response:
References were added (Lines 66 and 75) and additional references have been added to the revised Introduction.
Materials and Methods:
5. Authors should revise methods:
- Regarding application of protection motivation model for intention to take Covid 19 vaccine of general public, more relevant articles should be included: especially studies on “uptake of influenza vaccine” and related variables, and Asian countries and religious and cultural issues.
Authors’ response:
Studies on “uptake of influenza vaccine” and related variables were added (Lines 102-106).
6. Authors should state the reasons for applying protection motivation theory to uptake of Covid -19 vaccination and related factors (why excluding safety? Only financing BARRIER?) with reviewing previous literature or observation. Additionally, authors should revise the last sentence…aims of this study...
Authors’ response:
We have made clear the reason for applying PMT in the ‘Introduction’ (Lines 44-120). Our study does not exclude safety. Vaccine safe is a key control variable (Lines 161-164). The last sentence has been revised (Lines 118-120).
7. Authors should revise the method section carefully and rewrite following section more logically to better understand the text: study design, setting, Sample size and sampling method, Inclusion and exclusion criteria.
- provide “how to determine calculate the sample size for this study.
- provide the details of survey instrument: how to develop the survey questionnaire; to examine content and face validation of the survey instrument, variables of this study, and etc.
Authors’ response:
The method section has been substantially revised (Line 122-173), with a detailed discussion of the sample size.
8. Data collection
- provide details of: who are surveyors; informed consent; response rate; study periods, etc.
Authors’ response:
The details of surveyors, informed consent and response rate etc. are stated in section ‘Sampling Method’ (Lines 123-139).
9. Data analysis
- provide details of: statistical analyses for this study. (rely on hypothesis?)
Authors’ response:
The details of statistical analyses and the PMT hypothesis have been stated in section ‘2.5 Statistical Analyses’ (Lines 171-173), and the underlying hypothesis specified in lines 54-61, 118-120 and 276-280. Figure 1 and the PMT also make clear the model that is estimated (Lines 62-84).
10.Result and Discussion
- Authors should revise the result and discussion section: considering appropriate existing and updated articles related to “protection motivation theory and vaccination”.
11.References
- Authors should include more articles related to applying protection motivation theory and related factors.
Authors’ response 10 and 11:
The result and discussion section have been revised, with additional PMT references. Lines 281-305 reference previous PMT researches.
Round 2
Reviewer 2 Report
Thank you for addressing the comments. The draft is improved but there are several points that needs to be addressed as stated below.
Overall:
- the manuscript should be carefully reviewed to eliminate grammatical errors.
lines 82-102: the authors should analyze the findings, instead of simply narrating previous study results.
Table 3: Is there a particular reason why the authors chose to include all variables in the regression model even if the variable was identified as insignificant from the chi-square analysis?
Discussion section:
Implication of the findings are not fully discussed.
- what may have influenced the regional differences in vaccination intention rate?
- why are certain variables associated/not associated with vaccination intention?
- how/why is your result similar/different from the previous findings?
Line 314-315:
“First, PMT was used to analyze…………….to receive the COVID-19 vaccine”:
- the authors should elaborate on why this is a strength of this study?
- what is the significance of using this framework?
Author Response
Thank you for addressing the comments. The draft is improved but there are several points that needs to be addressed as stated below.
Overall:
- the manuscript should be carefully reviewed to eliminate grammatical errors.
lines 82-102: the authors should analyze the findings, instead of simply narrating previous study results.
Authors’ response:
We thank Reviewer 2 for this comment. Following the Reviewer’s previous request to provide further references in the Introduction, we expanded the references in the Introduction to place our study in the context of previous studies. We analyzed these previous studies in the Discussion section on page 10 lines 281-304.
Table 3: Is there a particular reason why the authors chose to include all variables in the regression model even if the variable was identified as insignificant from the chi-square analysis?
Authors’ response:
According to previous studies, the insignificant variables in the chi-square analysis in this study could be potential determinants which could influence the COVID-19 vaccination intention. So, we put them in the regression model. Adding these insignificant chi-square variables also made our regression more powerful with a larger R2. (Table 3)
Discussion section:
Implication of the findings are not fully discussed.
- what may have influenced the regional differences in vaccination intention rate?
- why are certain variables associated/not associated with vaccination intention?
- how/why is your result similar/different from the previous findings?
Authors’ response:
There were no regional differences in our multivariate ordered logistic regression model. We addressed the lack of region significance on lines 290-292, but, given the regional variable insignificance, the variable was not discussed further. Where the control and PMT variables were similar to previous studies, we pointed out the similarities, for example on lines 282-286, 300-302, and 307-308. In the Discussion section, we expanded our discussion of our findings and previous research on lines 313-314 and lines 319-322.
Line 314-315:
“First, PMT was used to analyze…………….to receive the COVID-19 vaccine”:
- the authors should elaborate on why this is a strength of this study?
- what is the significance of using this framework?
Authors’ response:
The strength and significance of using PMT has been revised in lines 326-329. The strength of PMT is also addressed in lines 53-59.